# New Biological and Chemical Insights into Optimization of Chamomile Extracts by Using Artificial Neural Network (ANN) Model

**DOI:** 10.3390/plants12061211

**Published:** 2023-03-07

**Authors:** Aleksandra Cvetanović Kljakić, Miloš Radosavljević, Gokhan Zengin, Linlin Yan, Uroš Gašić, Predrag Kojić, Aleksandra Torbica, Miona Belović, Zoran Zeković

**Affiliations:** 1Faculty of Technology, University of Novi Sad, Bulevar Cara Lazara 1, 21000 Novi Sad, Serbia; 2Department of Biology, Science Faculty, Selcuk University, Konya 42130, Turkey; 3Institute of Chemical Industry of Forestry Products, Chinese Academy of Forestry, Nanjing 210042, China; 4Institute for Biological Research “Siniša Stanković”, National Institute of Republic of Serbia, University of Belgrade, Bulevar Despota Stefana 142, 11060 Belgrade, Serbia; 5Institute of Food Technology, University of Novi Sad, Bulevar Cara Lazara 1, 21000 Novi Sad, Serbia

**Keywords:** chamomile, microwave-assisted extraction, ANN, UHPLC-LTQ-Orbitrap, enyzme-inhibition activity, antioxidant activity

## Abstract

Chamomile is one of the most consumed medicinal plants worldwide. Various chamomile preparations are widely used in various branches of both traditional and modern pharmacy. However, in order to obtain an extract with a high content of the desired components, it is necessary to optimize key extraction parameters. In the present study, optimization of process parameters was performed using the artificial neural networks (ANN) model using a solid-to-solvent ratio, microwave power and time as inputs, while the outputs were the yield of the total phenolic compounds (TPC). Optimized extraction conditions were as follows: a solid-to-solvent ratio of 1:80, microwave power of 400 W, extraction time of 30 min. ANN predicted the content of the total phenolic compounds, which was later experimentally confirmed. The extract obtained under optimal conditions was characterized by rich composition and high biological activity. Additionally, chamomile extract showed promising properties as growth media for probiotics. The study could make a valuable scientific contribution to the application of modern statistical designs and modelling to improve extraction techniques.

## 1. Introduction

Chamomile (*Chamomilla recutita* L.) is an annual plant whose medicinal properties have been known to almost all civilizations of the world for centuries, and some ancient civilizations considered it as a sacred plant. Besides its healing properties, its mild effect on the body without toxicity is of paramount importance when it comes to using it as a tea and/or as a medicine for babies and adults [1]. Among all its uses, its use in the fight against gastrointestinal diseases is particularly important, and there is almost no such type of disease for which chamomile cannot be used as medicine or prevention. In addition, its role as an antidepressant, its effect against insomnia, its use for rinsing wounds, cuts, and rashes, have been noted [2]. Precisely because of the beneficial effect on the skin, there are numerous skin care preparations based on chamomile. Modern science has established its anti-inflammatory, antiphlogistic [3], anti-allergic [4], antibacterial [5,6,7], antispasmodic, antiseptic [8], as well as antioxidant activity through numerous in vitro and in vivo tests [9]. More than 100 different compounds make up its composition and contribute to its activity and wide use. Studies show that the correct choice of extraction technique, solvents, and extraction conditions can favor the content of individual components in the final extract, and that to some extent the content of certain bioactive components can be favored, which can increase the bioactivity of the final product [10]. 

Among modern extraction techniques, microwave-assisted extraction (MAE) plays a particularly important role in the laboratory and increasingly in industry. MAE is a modern technique of extraction that combines traditional (solid–liquid extraction) with microwave heating to achieve a higher process effect [1]. In addition, the extraction process can be enhanced by enzymes. In general, enzymes play a key role in extraction as biocatalysts of highly specific reactions under mild conditions that drive the reaction in the desired direction, in an environmentally friendly environment (water), which characterizes the technique as green. Enzymes can degrade or hydrolyze cell wall elements and membranes, allowing for the better release and more efficient extraction of bioactive molecules. Enzymes such as beta-glucosidase can hydrolyze the bound form of molecules (such as flavonoid glucosides) and further increase the levels of the free form of these molecules (aglycones). In this way, the amount of aglycones (which often can be more bioactive) is higher, thereby increasing the bioactivity of the final extract [9]. However, in order to lead the processes in the desired direction and to be economically viable, it is necessary to know exactly the connection and impact of all key extraction parameters and target compounds. 

In MAE, it is very hard to find a theoretical model that explains the connection between input and outputs variables. Nowadays, with the evolution of hardware, software consequently becomes more powerful and data driven modelling such as artificial neural networks (ANNs) find an everyday use in science. ANN, as a direct and fast analysis of experimental data, can serve as an adequate alternative to models based on phenomenological hypotheses. It is worth nothing that the ANN models have already been applied in the extraction processes [11,12,13]. New and powerful optimization techniques have been developed parallel with data-driven modelling. Therefore, in order to overcome the possible local/minimal/global problems, optimization was settled. The two most used solvers in global optimization are the particle swarm and genetic algorithm. Each solver has its own characteristics that lead to different solutions. In order to improve solver effectiveness, many parameters need to be adjusted and different search functions have to be tested, so there is still place for progress in this scientific field. Particle swarm is a population-based algorithm. It is very similar to the genetic algorithm. Kennedy and Eberhart [14] introduced the particle swarm algorithm. Later, Mezura-Montes and Coello [15] improved the algorithm. It started by creating initial particles and assigning them the initial velocity values. A group of the particles move in steps through an area. The algorithm at each step determines the objective function of each particle. It determines the lowest function value and the best particle location. When the particles move, the algorithm reevaluates. It chooses new velocities based on the current velocity and the best location of the particles. It then iteratively updates the velocities and locations. This proceeds until the algorithm reaches a stopping criterion and finds the global optima.

One of the goals of this study was to optimize the microwave-assisted extraction process combined with enzymes in order to obtain the highest yield of phenols. The greater performance of prediction models is still the first-line goal in academia and the industrial community, so the ANN was used to algorithmically map input and outputs variables. The ANN is a black box model, and as the fitting function for optimization problems, it can lead to a solution that represents the local minimums. To overcome this, in the last phase we used a global optimization to find the best operating parameters for the microwave extraction. In terms of increasing the added value of the extracts, the growth of potential probiotic bacteria on the obtained extract was also evaluated.

## 2. Results and Discussion

### 2.1. Extraction and Optimization of the Extraction via ANN

One of the crucial moments in the process of obtaining natural products with added value is the process of extraction. The choice of extraction technique as well as the choice of solvent can significantly influence the presence and concentration of bioactive substances in the final extract. In this work, chamomile extraction was performed via MAE with ethanol. Ethanol is suitable for extracting plant matter because it is labeled as GRAS and can easily penetrate the pores of the plant matrix. MAE combined with ethanol is considered a green technique that requires less extraction time compared to traditional techniques [10]. Microwave heating also increases the temperature of the mixture, resulting in a better extraction performance. However, when extracting thermolabile compounds, high temperatures can lead to the decomposition of the components. The microwave power in the microwave extraction must be chosen sufficiently to avoid too-high temperatures, which leads to the decomposition of heat-sensitive substances [16,17,18]. Within this work, the ANN analysis was applied with the aim of determining the optimal extraction conditions and to gain an insight into the detailed influence of the extraction parameters, as well as to define the optimal extraction conditions for obtaining polyphenol-rich extracts.

Weights and bias initial values could significantly influence the result of the ANN. Also, the number of hidden neurons influence the predictability of the ANN. Hence, a different ANN architecture was tried, and the best ANN showed a good predictability performance. The best fitting ANN topology consisted of three layers with neurons: the input layer had three neurons, the hidden layer had eight neurons, and the output layer had one neuron [19]. The quality of the fit with respect to the coefficient of determination (R2) and mean absolute error (MAE) is presented in Figure 1, showing the parity plot of the experimental and predicted TPC yield. The parity plot demonstrated the generalization ability of the developed ANN. The ANNs were trained on the 70% randomly sampled points and were tested on the 30% remaining points.

In the next step, a developed ANN was incorporated into the Yoon’s model in order to determine the relative influence of the process parameters. The obtained relative importance values and the standard deviations are presented in Figure 2. A low variability in RI %, about 5%, makes the explanation of the input influence acceptable. Figure 2 shows that solid-to-solvent is the most influential parameter. The TPC yield decreased when decreasing the solid-to-solvent ratio from 1:80 to 1:40. From our experiments, it is important to note that the solid-to-solvent ratio of 1:40 produced the smallest TPC yield up to 46 mg/g. Hence, this leads to the conclusion that the solid-to-solvent ratio should not be smaller than 1:60. The maximal value for a TPC yield, of 59.56 mg/g, was obtained when the time of extraction was the longest, at 40 min, and the maximal microwave power was 800 W. Additionally, a similar TPC yield of 57.32 mg/g was obtained at 800 W and 20 min. So, this brings the conclusion that there is no need for a high microwave power to perform a long extraction exposure, because that increases the energy consumption. Moreover, from Figure 2 it can be seen that the microwave power has the smallest influence on the TPC yield. This is experimentally confirmed; the second highest value for TPC yield was 58.435 mg/g, regardless of the minimal microwave power of 400 W.

The ANN used in this study represents the black box model, so standard optimization methods such as response surface methodology can be trapped into the local minimum. This was reason to use global search solvers such as the particle swarm algorithm. Therefore, in the final stage, the particle swarm optimization determined the following operating parameters: 400 W, 30 min, and a 1:80 solid-to-solvent ratio. The predicted TPC yield obtained with the above-mentioned conditions was 59.56 mg/g. These results have been experimentally verified.

### 2.2. Chemical Profiling

Chamomile extract obtained under previously defined optimal conditions was exposed to the chemical profiling analysis. The first spectrophotometric assays were applied to determine the total phenol and flavonoid content, and the obtained results are presented in Figure 3. As can be seen, the total phenol content was 55.21 mg GAE/g, which was very close to the predicted value, confirming the ANN model. Analysis of the total amount of flavonoids showed the content of this bioactive compounds to be 44.98 mg RE/g, which represents more than 80% of the total polyphenolic compounds in the extracts, showing that flavonoids are the dominant group of polyphenols presented in chamomile. 

Apart from the spectrophotometric analysis, an analysis of the polyphenolic profile of the extract obtained under the optimal extraction conditions was performed as well. For this purpose, the UHPLC-LTQ OrbiTrap MS technique was used and a total of 67 compounds were identified (Table 1).

Fifteen compounds were identified via a comparison with the available analytical standards, while the others were tentatively identified by the search for [M–H]− deprotonated molecules in combination with MS4 fragmentations. To facilitate the understanding of the obtained results, the identified compounds are divided into two groups: (1) phenolic acids and their derivatives (32 compounds) and (2) flavonoid glycosides and aglycones (35 compounds) (Table 1).

The most represented compounds from the group of phenolic acids and their derivatives were hexosides and quinic acid esters of hydroxycinnamic acids (p-coumaric, caffeic, and ferulic acids). A total of 22 hydroxycinnamates were identified, which is in line with previously published results [20]. The other 10 compounds were hydroxybenzoic acid derivatives. Among all compounds from this subgroup, it is interesting to single out compound 21 from Table 1. In the first stage of fragmentation (MS2 base peak) it gives 413 *m*/*z*, which can occur with a loss of 138 Da (mass of hydroxybenzoic acid). Further fragmentation (MS3 base peak) produces an ion at 137 *m*/*z*, which also corresponds to the mass of deprotonated hydroxybenzoic acid. The difference between 413 and 137 *m*/*z* gives a neutral loss of 276 *m*/*z*, which may correspond to deprotonated vicianose (arabinopyranosyl-glucopyranose) or sambubiose (xylosyl-glucopyranose). All these facts show that this compound probably contains two molecules of hydroxybenzoic acid: one pentose and one hexose unit.

Considering the identified flavonoid glycosides and aglycones, the most abundant were apigenin derivatives, as was expected [21]. Nine compounds were confirmed via a comparison with the standards (37, 42–44, 57, 59, 63, 65, and 66) and the others were tentatively identified via examination of the MS spectra. Some flavonoids specific for chamomile, such as 6-methoxyapigenin or hispidulin, 6-methoxyluteolin or nepetin [22], 6-hydroxyquercetin or quercetagetin [23], and 6-methoxyquercetin or patuletin [24], were also found in this study. Compound 50 was marked as an apigenin derivative, because its exact structure could not be suggested. Based on the isotopic composition and exact mass (569.12927 *m*/*z*), the chemical formula of this compound (C28H26O13) can be calculated. In the first fragmentation step, this compound loses the 300 Da and thus the MS2 base peak at 269 *m*/*z* was formed. Further MS3 and MS4 fragmentation confirms the presence of apigenin as one part of this compound, because it is in accordance with the fragmentation of compound 41 (apigenin 7-O-glucoside).

By using standards, quantification was performed for the 15 compounds, and the obtained results are presented in Table 2. Among the phenolic acids, p-hydroxybenzoic was presented in the highest concentration (1.619 mg/L), which was in accordance with the literature data, and which suggests that it is the most abundant compared to other phenolic acids present in chamomile. Regarding the flavonoids, as was expected, apigenin and apigenin-7-O-glucoside were the dominant compounds. Generally, apigenin is considered as one of the leading constituents of chamomile and responsible for many of its biological activities [9]. In addition to apigenin, the concentration of quercetin, which is considered an extremely potent biological molecule, was also very high (0.861 mg/L), and is of notable importance for the overall activity of the extracts. Namely, this compound is known as one of the most potent ingredients of many plants and possesses antioxidant, antimicrobial, antitumor, anti-inflammatory, and immunosuppressive effects, among many others [25]. Kaempferol was present in the extract with a concentration of 0.358 mg/L. The presence of kaempferol can have a significant role in the biological activity of chamomile extracts due to the wide range of its biological activities. This compound is used for reducing the risk of chronic diseases, such as cancer, liver injury, obesity, and diabetes [26,27]. Because of its anti-inflammatory potential, kaempferol is used for different inflammation-induced diseases, such as intervertebral disc degeneration and colitis, post-menopausal bone loss, and acute lung injury [28].

### 2.3. Biological Activity

#### 2.3.1. Antioxidant Activity

When it comes to the biological activity of natural products, one of the fundamental activities is antioxidant activity. The very abundant presence of this activity is the basis for other biological activities such as antimicrobial and cardioprotective activities, etc. There are generally two ways in which the precursor compounds exert their antioxidant activity. Namely, they can prevent the oxidation process or neutralize free radicals. For this very reason, when determining the antioxidant capacity of natural products, it is desirable to use several different tests based on different mechanisms of action. In this study, the antioxidant capacity of the chamomile extract was tested using six different tests, which gives a more realistic insight into the potential of the obtained extract to be used as a source of antioxidants (Table 3). Two applied antiradical assays confirmed that the chamomile extract possessed a high potential to neutralize DPPH (60.24 mg TE/g) and ABTS (126.92 mg TE/g) free radicals. By applying two reduction assays, the high reduction ability of the extract was confirmed. Namely, by using CUPRAC (123.12 mgTE/g) and FRAP (95.37 mg TE/g) tests, the chamomile-optimized extract showed its high reduction capability. Results obtained for the chelating metal test (21.34 mg EDTAE/g) express higher activity than the activity of other plant species obtained via the same extraction technique. Namely, in our previous published data, Origanum vulgare (6.49 mg EDTAE/g) [29] and Tanacetum parthenium (10.83 mg EDTAE/g) [30] extracts obtained via the MAE extraction technique express lower antioxidant potential in comparison with MAE chamomile extract obtained in this study.

#### 2.3.2. Enzyme-Inhibitory Activity

Enzymes are important therapeutic targets for the treatment of global health issues such as Alzheimer’s disease, diabetes, and obesity. Some enzymes can be targeted for this purpose, and their inhibition can reduce disease symptoms. For instance, amylase and glucosidase are known as anti-diabetic enzymes, and their inhibition can help diabetic patients control their blood glucose levels [31]. Several compounds have been synthesized in the pharmaceutical industry as enzyme inhibitors, but the majority of them have undesirable side effects. Thus, there is a great deal of interest in replacing natural enzyme inhibitors with synthetic ones. The enzyme inhibitory properties of the optimized chamomile extract against acetylcholinesterase (AChE), amylase, and glucosidase were investigated. The results are tabulated in Table 3. The optimized extracts exhibited inhibitory properties on the tested enzymes (AChE: 0.85 mg of GALAE/g; amylase: 0.18 mmol of ACAE/g; and glucosidase: 13.11 mmol of ACAE/g). The observed enzyme inhibitory abilities can be explained by the presence of some compounds in the optimized extract. For instance, several studies have been reported that apigenin has great potential as one of natural inhibitors against AChE [32], amylase, and glucosidase [33]. In addition to apigenin, the presence of caffeic and chlorogenic acids can be attributed to observed abilities [34,35]. Our results are comparable to the reported values in the literature. For example, in our previous study [36], the chamomile extracts from subcritical water extracts under different pressures exhibited inhibitory effects on amylase (0.44–0.46 mmol ACAE/g) and glucosidase (2.42–3.60 mmol ACAE/g). In another study [37], the amylase- and glucosidase-inhibitory effects of the optimized extract for apigenin isolation were 0.45 mmol of ACAE/g and 2.54 mmol of ACAE/g, respectively. The values were also reported as 0.94 mmol of ACAE/g and 3.24 mmol of ACAE/g for autofermented chamomile ligulate flowers [38]. Taken together, chamomile extracts have significant enzyme inhibitory properties on anti-diabetic enzymes. The results presented could contribute significantly to the design of natural enzyme inhibitors for pharmaceutical applications.

### 2.4. Evaluation of Potential of MAE for the Growth of La. rhamnosus ATCC 7469

Research shows that plant extracts that are rich in biologically active compounds in combination with probiotics show a significant effect against inflammatory diseases and disorders. Namely, the synergistic action of bioactives and probiotics express a promising potential in the amendment of inflammatory diseases and disorders [39,40]. From those reasons, the influence of chamomile extract on the growth of *La. rhamnosus* was determined.

As previously stated, since the MAE did not exhibit a significant antimicrobial effect, especially on bacteria (*La. rhamnosus* ATCC 7469 was also included in microwell dilution method), the potential of MAE and MAEC for growth of *La. rhamnosus* was investigated. The reducing sugar utilization after 24 h for MAE and MAEC was 68.4 and 82.4%, respectively, and at this point the increase in viability stopped. The initial viability (at 0 h) in all experiments was approximately 8 log CFU/mL, whereas after 24 h the significant increase in viability was observed. The final viability for MAE and MAEC was 9.2 log CFU/mL and 9.7 log CFU/mL, respectively, resulting in significant increases of 15 and 20%. The findings suggest that *La. rhamnosus* could be successfully grown on this type of chamomile extract. However, the production of lactic acid was not significant (data not shown), suggesting that MAE and MAEC are not complete mediums for lactic acid fermentation, since the reducing sugars concentration is relatively low for efficient lactic acid fermentation. Nevertheless, the MAE and MAEC could be used for enhancement (stimulation) of lactic acid bacteria growth and/or formulation of synergetic bioactive compound-rich probiotic preparations. Marhamatizadeh et al. [41] successfully used *Chamomile Essence* for enhancement of *Bifidobacterium* and *Lactobacillus* strains in milk and yoghurt. *Echium amoenum* extract was also successfully used for enhancement of the growth of *Bifidobacterium* and *Lactobacillus* strains in kefir [42]. Similar results were observed by Dimofte et al. [43] when fermentation media for *Weissella confuse* PP29 was enriched with deferment concentrations of anthocyanins from Hibiscus sabdariffa L. The fermentation media rich in anthocyanins (high concentrations) stimulated the growth of *Weissella confuse* and biosynthesis of exopolysaccharide, thus leading to augmented probiotic and prebiotic properties. Martinelli et al. [44] determined (during a clinical trial) that administration of *M. chamomilla* L., *M. officinalis* L., tyndallized *L. acidophilus* (HA122), and *L. reuteri* DSM 17938 was more effective than simethicone on an infant colic. Synergetic action of probiotic bacteria with herbal extracts (Chamomile (*Matricaria chamomilla*), fennel (*Foeniculum vulgare*), peppermint (*Mentha piperita*), and thyme (*Thymus vulgaris*) were proven as efficient antibacterial agents against *Escherichia coli*, *Salmonella typhimurium*, and *Salmonella* [45].

The connection of medical plants and probiotics is well known in the traditional medicine of many countries [46]. The current interest, with constantly increasing relevance in relation to healthy supplements and diets, indicates the potential of novel delivery systems that combine probiotics with plant extracts rich in bioactive compounds. This simultaneous and synergistic delivery system could be significantly beneficial for health. In this sense, research on the complementary benefits of herbal extracts and probiotics, as well research on the potential enrichment of food via the incorporation of these bioactive compounds and improvements in probiotic survivability, is highly valuable [47].

## 3. Material and Methods

### 3.1. Chemicals

Acetonitrile and formic acid (both of them LC/MS grade) were purchased from Merck (Darmstadt, Germany). Standards of phenolic compounds (gallic acid, protocatechuic acid, p-hydroxybenzoic acid, gentisic acid, caffeic acid, p-coumaric acid, apigenin 7-O-glucoside, luteolin, apigenin, chrysoeriol, quercetin 3-O-galactoside, isorhamnetin 3-O-glucoside, quercetin, and kaempferol) were supplied by Sigma Aldrich (Steinheim, Germany). The 3.5-dinitrosalicylic acid was also supplied by Sigma Aldrich (Steinheim, Germany). Syringe filters (25 mm, nylon membrane, 0.45 µm) were purchased from PSI Lab (Belgrade, Serbia). MRS broth and agar were supplied by HiMedia (Mumbai, India). 

### 3.2. Plant Material

For the purpose of this study, ligulate flowers of chamomile (CLF) were used. Plant material was obtained from the Institute of Field and Vegetable Crops, Novi Sad, Serbia. The collected material was firstly dried (40 °C), and then CLF were separated from the tubular parts of the flower by sieving. Prior to extraction, Dried CLF were treated using a sodium–acetate buffer under the conditions which were optimal for the activity of β-glucosidase [9]. The conditions were kept constant for 72 h and afterwards dried at room temperature for five days. Such plant material was further extracted via ethanol by applying the MAE extraction technique.

### 3.3. Extraction, Experimental Design, and ANN Optimization

Microwave-assisted extraction (MAE) was performed in a modified domestic microwave oven (Figure 4). In order to define the best conditions for obtaining extract with the maximum phenol content, a 33 full factorial design was chosen. Plant material was mixed with different proportions of ethanol (1:40, 1:60, 1:80) and exposed to microwave irradiation of different powers (400, 600, 800 W) for different periods of time (20, 30, 40 min). These three parameters (solid-to-solvent ratio, microwave power, and time) represented the inputs and were used to feed the artificial neural network. The experimental design with 27 runs is shown in the Table 4. The feed-forward back propagation ANN was used to predict the experimental values for phenol yield outputs. 

The transfer function was the linear (purelin) at the output layer and the tangent sigmoid function (tansig) at the hidden layer. The best predictive ability was achieved via the Levenberg–Marquad solver. Successful creation of the ANNs and obtained weight matrices provide the determination of the relative importance (RI) of the input values and its effect on the phenols yield by using a partitioning methodology. In this study, the following Yoon’s equation was used [48]:
$$RI_{ij}\;\left[\%\right]=\frac{\sum_{k=0}^{n}\left(w_{ik}\;w_{kj}\right)}{\sum_{i=0}^{m}abs\;\sum_{k=0}^{n}\left(w_{ik}\;w_{kj}\right)}\;100{\%}_{\;}$$
where *RI_ij_* is the relative importance of the *i*th input variable on the *j*th output, *w_ik_* is the weight between the *i*th input and the *k*th hidden neuron, and *w_kj_* is the weight between the *k*th hidden neuron and the *j*th output. This analysis proved that the decision variables have a conflicting influence on the performance parameters. Thus, to achieve the maximal microwave extraction yield, a global optimization with the particle swarm solver was used. All data analysis was performed using MATLAB 2016b.

### 3.4. Chemical Analysis of Extracts

#### 3.4.1. Assays for Total Phenolic and Flavonoid Content

The content of two major groups of bioactive compounds (phenols–TPC and flavonoids–TFC) in obtained extracts was determined spectrophotometrically using appropriate Folin–Ciocalteu and aluminium chloride methods [49]. Expression of obtained results was performed using equivalent of standards–gallic acid (in thw case of TPC) and rutin (in the case of TFC).

#### 3.4.2. UHPLC-LTQ OrbiTrap MS Analysis of Polyphenolic Compounds

Separation and tentative identification of bioactive compounds in tested chamomile extract were performed by using an ultra-high-pressure liquid chromatography (UHPLC) system (Accela 600, ThermoFisher Scientific, Bremen, Germany) coupled to linear ion trap/orbitrap mass spectrometer (LTQ OrbiTrap MS) from ThermoFisher Scientific, Bremen, Germany). All the details about chromatographic separation settings [50], electrospray ionization (ESI) source, and MS detector parameters [51] were previously described.

Identification of some compounds was achieved by comparison with the available standard, while other compounds were tentatively identified via high resolution mass spectrometry (HRMS) in combination with multistage mass spectrometry (MS^n^). The available literature about analyses of bioactive compounds in various chamomile species was used to confirm the identified compounds [20,21,52,53,54,55].

### 3.5. Determination of Biological Activity of Extracts

To provide comprehensive insights into the biopotential of the extracts, the antioxidant, anti-α-amylase, anti-α-glucosidase, and anti-cholinesterase activities were determined. Estimation of the anti-enzymatic activity of the extracts was conducted via in vitro assays previously described by Uysal et al. [56]. The data obtained via these assays were given as reference inhibitors equivalents: galantamine (GALAE) for acetylcholinesterase (AChE), acarbose (ACAE) for α-amylase and α-glucosidase. Measurements of the antioxidant and free radical-scavenging properties of the extracts, ferric reducing antioxidant power (FRAP), 2,2′-azino-bis(3-ethylbenzothiazoline-6-sulphonic acid) (ABTS), cupric reducing antioxidant capacity (CUPRAC), 2,2-diphenyl-1-picrylhydrazyl (DPPH), metal chelating, and phosphomolybdenum were performed. The data were given as reference compounds (Trolox (TE) and ethylenediaminetetraacetic (EDTAE) equivalents). Detailed description of applied assays was given previously [57,58,59].

### 3.6. Evaluation of Chamomile Potential for the Growth of La. rhamnosus ATCC 7469

#### 3.6.1. Preparation of Growth Media and Inoculum, and Bacteria Growth Conditions

Since the MAE did not exert a significant antimicrobial effect, especially towards bacteria (preliminary research), the MAE was used as potential growth medium for potential prebiotic *Lacticaseibacillus rhamnosus* [60] (previously known as *Lactobacillus rhamnosus*) [61]. Sterilized MAE without and with prior concentration (MAEC, up to 50% of the original volume) were used as potential growth media of lactic acid bacteria. MAE and MAEC contained 0.7% and 1.3% of reducing sugars, respectively.

The stock culture of the homofermentative lactic acid (LA) strain *La. rhamnosus* (ATCC, Rockville, MD, USA) was activated as previously described [62]. The inoculums for free cell fermentations were prepared by transferring 3 mL of activated culture into 60 mL of MRS (de Man, Rogosa and Sharpe) broth. Inoculums for immobilization experiments were prepared by adding 7 mL of activated culture into 120 mL of MRS broth. In order to achieve high LAB viability (9 log CFU/mL), all inoculums were incubated at 37 °C for 24 h.

All LA fermentations were performed with shaking (150 rpm, Biosan shaking bath model ES-20, Biosan Ltd., Riga, Latvia). The experiments were performed in triplicates. The fermentations were performed in 300 mL Erlenmayer flasks with 200 mL of MAE or MAEC media. The fermentation was initiated by the addition of inoculum (5% *v*/*v*) and conducted at 37 °C. In order to neutralize the produced LA, sterile calcium carbonate (CaCO_3_, 0.5% (*w*/*v*)) was added to all fermentation media prior to inoculation.

#### 3.6.2. Determination of Reducing Sugars Concentration and Viability of *La. rhamnosus* Cells

Reducing sugar concentration, calculated as glucose, was determined via the 3.5-dinitrosalicylic acid method [63] using UV/VIS Spectrometer (UV-1800, Shimadzu, Kyoto, Japan). A calibration curve was set at 570 nm using standard glucose solutions. A number of viable *La. rhamnosus* cells were determined using a pour plating method. Microaerophilic conditions were maintained during incubation in Petri plates using a double MRS agar medium layer. Samples were incubated for 48 h at 37 °C. The total viable cell number was expressed as log CFU/mL. All chemicals used in experiments were of analytical and microbiological grade.

## 4. Conclusions

The aim of this study was to use an environmentally friendly and sustainable extraction technique to isolate bioactive compounds from *Matricaria chamomile*. With this in mind, we selected ethanol and microwave extraction and optimized the procedure by applying the ANN methodology. To provide sufficient input experimental data to train the ANN, a 33 full factorial design for 3 factors was chosen. An experimental design with 27 runs combining extraction time, microwave irradiation power as well as solid-to-solvent ratio was therefore conducted. These data were used to feed the artificial neural network that will predict the desired outputs (total phenol content). The optimized conditions were 400 W, 30 min, and 1:80. The predicted TPC yield obtained with such conditions was 59.56 mg/g, and these results have been verified experimentally. The optimized chamomile extracts were rich in total phenols (55.21 mg CAE/g) and total flavonoids (44.98 mg RE/g). LC-MS analysis confirmed the presence of 67 polyphenolic compounds among which apigenin and apigenin-7-O-glucoside were the dominant. In addition, the bioactivity of the optimized extract was evaluated, showing a promising ability to inhibit antioxidants and glucosidase. The results could be valuable for further industrial applications where chamomile is used as a source of natural active ingredients such as antioxidants, antimicrobials, or enzyme inhibitors.

## Figures and Tables

**Figure 1 plants-12-01211-f001:**
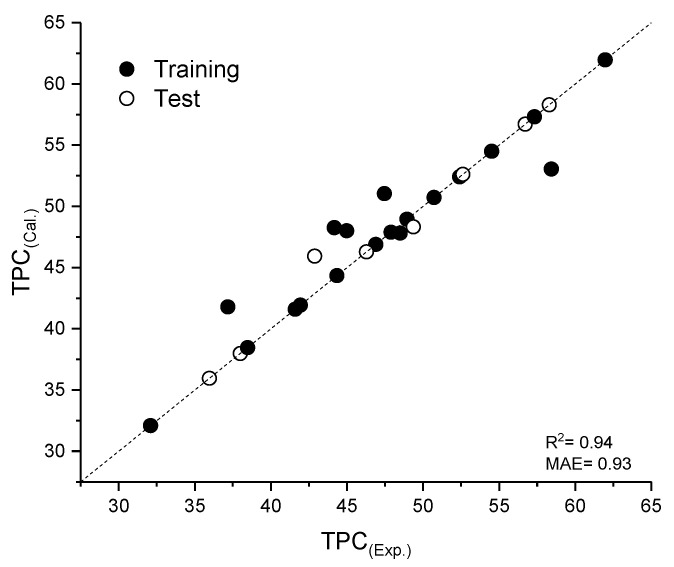
Parity plot of the experimental and predicted TPC yield.

**Figure 2 plants-12-01211-f002:**
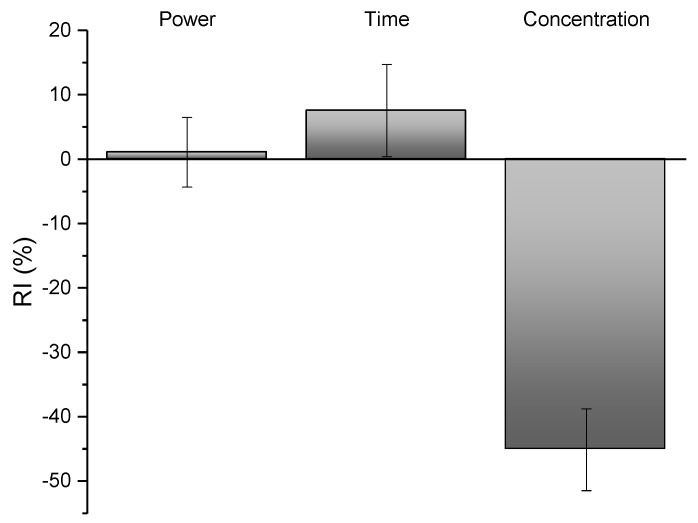
Relative importance of independent variables on the TPC.

**Figure 3 plants-12-01211-f003:**
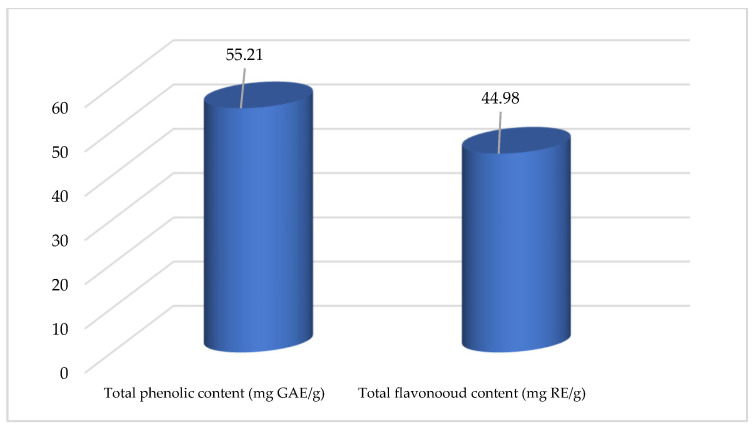
Total phenolics and flavonoids content in optimized chamomile extract.

**Figure 4 plants-12-01211-f004:**
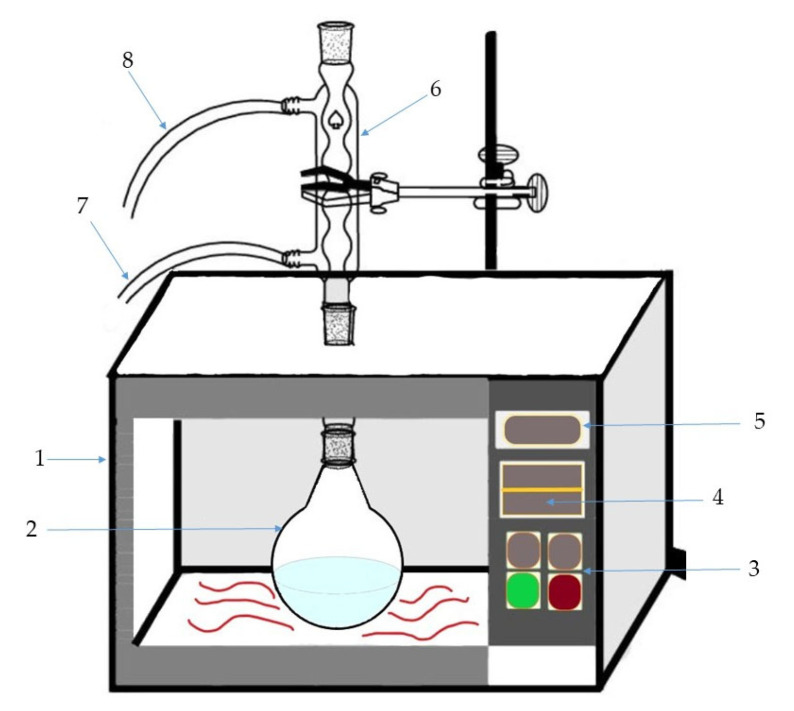
Schematic diagram of the microwave-assisted distillation apparatus 1. microwave oven, 2. glass flask, 3. irradiation power controller, 4. time controller, 5. display, 6. condenser, 7. inlet water, 8. outlet water.

**Table 1 plants-12-01211-t001:** Compounds identified in chamomile extract.

No	*t*_R_, Min	Compound Name	Molecular Formula, [M − H]^–^	Calculated Mass, [M − H]^–^	Exact Mass, [M − H]^–^	Δ ppm	MS^2^ Fragments, (% Base Peak)	MS^3^ Fragments, (% Base Peak)	MS^4^ Fragments, (% Base Peak)
	** *Phenolic acids and their derivatives* **								
**1**	1.95	**Gallic acid hexoside isomer 1**	C_13_H_15_O_10_^–^	331.06707	331.06680	0.82	211(15), 193(80), 175(30), **169**(100), 151(50)	**151**(100)	110(10), 97(30), 81(100), 53(30)
**2**	2.39	**Gallic acid *^a^***	C_7_H_5_O_5_^–^	169.01425	169.01385	2.37	**125**(100)	**107**(100)	−
**3**	2.69	**Dihydroxybenzoic acid hexoside isomer 1**	C_13_H_15_O_9_^–^	315.07216	315.07206	0.32	**153**(100), 152(50), 109(15), 108(10)	**109**(100)	123(25), 109(10), 85(10), 81(100)
**4**	3.13	**Gallic acid hexoside isomer 2**	C_13_H_15_O_10_^–^	331.06707	331.06702	0.15	**313**(100), 169(25), 168(90), 151(10), 125(25)	193(50), **151**(100), 125(80)	123(100), 107(90), 95(65)
**5**	3.82	**Dihydroxybenzoic acid hexoside isomer 2**	C_13_H_15_O_9_^–^	315.07216	315.07121	3.02	**153**(100), 135(10), 109(10)	**135**(100), 109(50)	91(100)
**6**	4.32	**Dihydroxybenzoic acid hexoside isomer 3**	C_13_H_15_O_9_^–^	315.07216	315.07169	1.49	**153**(100), 109(10)	**135**(100), 109(50)	91(100)
**7**	4.51	**Protocatechuic acid *^a^***	C_7_H_5_O_4_^–^	153.01933	153.01872	3.99	**109**(100)	81(60), 80(50), 67(30), **65**(100)	−
**8**	4.38	**Caffeoylquinic acid hexoside isomer 1**	C_22_H_27_O_14_^–^	515.14008	515.13928	1.55	353(80), 341(5), 323(10), **191**(100), 179(5)	173(65), 127(90), 111(50), 93(55), **85**(100)	−
**9**	4.61	**3-*O*-Caffeoylquinic acid**	C_16_H_17_O_9_^–^	353.08781	353.08676	2.97	**191**(100), 179(35), 135(10)	173(75), **127**(100), 111(40), 93(60), 85(90)	109(30), 99(40), 85(100)
**10**	4.68	**Caffeic acid hexoside isomer 1**	C_15_H_17_O_9_^–^	341.08781	341.08716	1.91	191(10), **179**(100), 135(10)	**135**(100)	135(100), 107(50)
**11**	4.88	**Caffeoylquinic acid hexoside isomer 2**	C_22_H_27_O_14_^–^	515.14008	515.13928	1.55	353(15), 341(15), **323**(100), 191(25), 179(5)	**161**(100), 133(5)	133(100), 117(20)
**12**	5.17	**Caffeic acid hexoside isomer 2**	C_15_H_17_O_9_^–^	341.08781	341.08731	1.47	**179**(100), 135(10)	**135**(100)	107(100), 79(20)
**13**	5.22	**Ferulic acid hexosylhexoside**	C_22_H_29_O_14_^–^	517.15628	517.15466	3.13	221(25), **193**(100), 179(25), 161(10), 149(20)	**149**(100)	134(100)
**14**	5.30	**4-*O*-Caffeoylquinic acid**	C_16_H_17_O_9_^–^	353.08781	353.08749	0.91	223(20), 191(50), 179(60), **173**(100), 135(10)	115(20), 111(50), **93**(100), 71(20)	−
**15**	5.32	**Coumaric acid hexoside**	C_15_H_17_O_8_^–^	325.09289	325.09283	0.18	**163**(100), 119(10)	**119**(100)	−
**16**	5.42	***p*-Hydroxybenzoic acid *^a^***	C_7_H_5_O_3_^–^	137.02442	137.02420	1.61	109(10), **93**(100)	**66**(100)	−
**17**	5.46	**Gentisic acid *^a^***	C_7_H_5_O_4_^–^	153.01933	153.01895	2.48	**109**(100)	95(10), **81**(100), 65(35)	−
**18**	5.73	**Ferulic acid hexoside isomer 1**	C_16_H_19_O_9_^–^	355.10346	355.10187	4.48	**193**(100), 149(25)	**149**(100)	133(100)
**19**	5.78	**3-*O*-Feruoylquinic acid**	C_17_H_19_O_9_^–^	367.10346	367.10263	2.26	**193**(100), 178(5), 173(5), 134(10)	178(90), 149(40), **134**(100)	106(100)
**20**	5.82	**Caffeic acid *^a^***	C_9_H_7_O_4_^–^	179.03498	179.03459	2.18	**135**(100)	135(60), 117(15), **107**(100), 91(55), 79(15)	−
**21**	5.86	**Hydroxybenzoic acid derivative**	C_25_H_27_O_14_^–^	551.14063	551.13977	1.56	431(10), **413**(100)	281(5), 179(5), **137**(100)	93(100)
**22**	5.94	**5-*O*-*p*-Coumaroylquinic acid**	C_16_H_17_O_8_^–^	337.09289	337.09268	0.62	**191**(100), 179(5), 163(10)	173(75), **127**(100), 111(40), 93(60), 85(90)	109(30), 99(40), 85(100)
**23**	6.39	**4-*O*-Feruoylquinic acid isomer 1**	C_17_H_19_O_9_^–^	367.10346	367.10275	1.93	193(5), **173**(100), 155(5), 111(5)	155(15), 111(40), 93(100), 71(10)	−
**24**	6.46	**Ferulic acid acetylhexoside isomer 1**	C_18_H_21_O_10_^–^	397.11402	397.11346	1.41	**193**(100), 149(30), 134(10)	**149**(100)	134(100)
**25**	6.51	**Ferulic acid hexoside isomer 2**	C_16_H_19_O_9_^–^	355.10346	355.10306	1.13	**193**(100), 149(10)	**149**(100)	133(100)
**26**	6.64	**4-*O*-Feruoylquinic acid isomer 2**	C_17_H_19_O_9_^–^	367.10346	367.10294	1.42	193(5), 179(10), **173**(100), 155(5), 111(10)	155(15), 111(40), **93**(100), 71(10)	−
**27**	6.72	***p*-Coumaric acid *^a^***	C_9_H_7_O_3_^–^	163.04007	163.04006	0.06	**119**(100)	119(60), 101(20), 93(25), **91**(100), 72(10)	−
**28**	6.94	**Dicaffeoylquinic acid isomer 1**	C_25_H_23_O_12_^–^	515.11950	515.11749	3.90	**353**(100), 355(20), 299(10), 191(10), 179(15)	191(40), 179(70), **173**(100), 135(10)	155(10), 111(50), **93**(100), 71(10)
**29**	7.09	**Ferulic acid acetylhexoside isomer 2**	C_18_H_21_O_10_^–^	397.11402	397.11400	0.05	**193**(100), 149(30), 134(10)	**149**(100)	134(100)
**30**	7.16	**Dicaffeoylquinic acid isomer 2**	C_25_H_23_O_12_^–^	515.11950	515.11780	3.30	**353**(100)	**191**(100), 179(30), 173(5), 135(10)	173(65), 127(90), 111(50), 93(55), 85(100)
**31**	7.33	**Dicaffeoylquinic acid isomer 3**	C_25_H_23_O_12_^–^	515.11950	515.11874	1.48	**353**(100), 355(5), 317(5), 299(10), 203(5)	191(30), 179(60), **173**(100), 135(10)	155(20), 111(60), 93(100), 71(20)
**32**	8.05	**Ferulic acid**	C_10_H_9_O_4_^–^	193.05063	193.04985	4.04	178(70), **149**(100), 134(40)	**134**(100)	−
	** *Flavonoid aglycones and glycosides* **								
**33**	5.62	**Apigenin 6,8-di-*C*-hexoside**	C_27_H_29_O_15_^–^	593.15119	593.15002	1.97	533(10), 503(30), **473**(100), 383(20), 353(35)	383(20), **353**(100)	275(20), 265(60), 249(100), 221(35), 173(80)
**34**	6.18	**6-Hydroxyquercetin 7-*O*-hexoside**	C_21_H_19_O_13_^–^	479.08311	479.08176	2.82	318(10), **317**(100)	299(40), **271**(100), 167(75), 139(45)	243(100), 227(30), 215(20), 199(50)
**35**	6.65	**Quercetin 3-*O*-galactoside *^a^***	C_21_H_19_O_12_^–^	463.08820	463.08753	1.45	**301**(100), 300(30)	273(25), 257(20), **179**(100), 151(75)	151(100)
**36**	6.74	**Luteolin 7-*O*-hexoside**	C_21_H_19_O_11_^–^	447.09329	447.09262	1.50	286(10), **285**(100)	257(30), **241**(100), 217(75), 199(85), 175(95)	241(5), 226(15), 213(30), 197(100)
**37**	6.79	**6-Methoxyquercetin 7-*O*-hexoside**	C_22_H_21_O_13_^–^	493.09876	493.09772	2.11	477(20), 373(10), **331**(100), 323(30), 316(5)	**316**(100), 209(5), 181(5), 166(5)	287(100), 271(60), 194(40), 166(70), 151(5)
**38**	6.92	**Apigenin 7-*O*-(6”-rhamnosyl)hexoside**	C_27_H_29_O_14_^–^	577.15628	577.15576	0.90	270(10), **269**(100)	**225**(100), 201(20), 197(30), 183(30), 149(25)	225(5), 210(10), 197(100), 181(50), 169(40)
**39**	7.07	**6-Methoxyapigenin 7-*O*-(6”-rhamnosyl)hexoside**	C_28_H_31_O_15_^–^	607.16684	607.16656	0.46	300(15), **299**(100), 284(5)	**284**(100)	284(30), 256(100), 239(5), 227(15), 211(10)
**40**	7.22	**Isorhamnetin 3-*O*-glucoside *^a^***	C_22_H_21_O_12_^–^	477.10385	477.10278	2.24	357(20), 315(50), **314**(100), 300(5), 299(5)	300(30), **285**(100), 271(75), 257(10), 243(25)	270(100)
**41**	7.27	**Apigenin 7-*O*-glucoside *^a^***	C_21_H_19_O_10_^−^	431.09837	431.09811	0.60	270(10), **269**(100)	**225**(100), 201(15), 197(20), 183(30), 149(30)	210(15), 197(80), 183(100), 181(30), 169(30)
**42**	7.30	**Naringenin *^a^***	C_15_H_11_O_5_^−^	271.06120	271.06113	0.26	177(10), **151**(100)	**107**(100)	65(100)
**43**	7.38	**Isorhamnetin 7-*O*-hexoside**	C_22_H_21_O_12_^–^	477.10385	477.10385	0.00	462(5), 357(10), 316(10), **315**(100), 300(5)	**300**(100), 285(5), 151(10)	283(20), 272(65), 271(70), 227(30), 151(100)
**44**	7.43	**6-Methoxyapigenin 7-*O*-hexoside**	C_22_H_21_O_11_^−^	461.10894	461.10834	1.30	446(80), 341(10), **299**(100), 284(20)	**284**(100)	284(30), 256(100), 239(5), 227(15), 211(10)
**45**	7.51	**6-Methoxyquercetin 7-*O*-(6”-caffeoyl)hexoside**	C_31_H_27_O_16_^–^	655.13046	655.13043	0.05	533(15), 505(10), **331**(100), 323(20), 316(30)	**316**(100), 209(5), 181(5), 166(5)	287(100), 271(60), 194(40), 166(70), 151(5)
**46**	7.54	**Isorhamnetin 7-*O*-(6”-acetyl)hexoside**	C_24_H_23_O_13_^–^	519.11441	519.11383	1.12	357(5), 316(10), **315**(100), 300(5), 285(5)	**300**(100), 287(5), 272(10)	272(30), 271(100), 255(50)
**47**	7.81	**Apigenin 7-*O*-acetylhexoside isomer 1**	C_23_H_21_O_11_^–^	473.10894	473.10782	2.37	413(15), 311(10), 270(15), **269**(100), 268(60)	**225**(100), 201(25), 197(35), 183(30), 149(40)	210(20), 197(100), 183(30), 181(60), 169(30)
**48**	8.15	**Apigenin 7-*O*-acetylhexoside isomer 2**	C_23_H_21_O_11_^–^	473.10894	473.10699	4.12	413(30), 311(10), 270(10), **269**(100), 268(40)	**225**(100), 201(30), 197(30), 183(30), 149(35)	207(10), 197(100), 183(30), 181(70), 169(40)
**49**	8.20	**Apigenin 7-*O*-(6”-caffeoyl)hexoside**	C_27_H_29_O_15_^–^	593.13006	593.12848	2.66	323(95), **269**(100), 221(15), 179(10)	**225**(100), 201(30), 183(10), 151(20), 149(35)	225(10), 208(10), 197(40), 181(100), 169(15)
**50**	8.38	**Apigenin derivative**	C_28_H_25_O_13_^–^	569.13006	569.12927	1.39	270(10), **269**(100)	**225**(100), 201(20), 197(30), 183(30), 149(25)	225(5), 210(10), 197(100), 181(50), 169(40)
**51**	8.50	**Apigenin 7-*O*-acetylhexoside isomer 3**	C_23_H_21_O_11_^–^	473.10894	473.10726	3.55	413(5), 311(10), 270(15), **269**(100), 268(50)	**225**(100), 201(30), 197(25), 183(25), 149(30)	197(70), 183(50), 181(100), 169(30)
**52**	8.55	**Apigenin 7-*O*-diacetylhexoside isomer 1**	C_25_H_23_O_12_^–^	515.11950	515.11713	4.60	455(30), 431(5), 413(10), 311(15), **269**(100)	**225**(100), 201(25), 197(25), 183(25), 149(30)	210(10), 197(100), 183(30), 181(50), 169(40)
**53**	8.63	**6-Methoxyapigenin 7-*O*-(6”acetyl)hexoside**	C_24_H_23_O_12_^–^	503.11950	503.11816	2.66	**488**(100), 299(20), 284(10)	429(10), 327(10), 313(40), **283**(100), 255(30)	255(100)
**54**	8.68	**6-Methoxyapigenin**	C_16_H_11_O_6_^–^	299.05611	299.05569	1.40	285(15), **284**(100)	**256**(100)	238(30), 228(70), 211(60), 188(100)
**55**	8.67	**Luteolin *^a^***	C_15_H_9_O_6_^−^	285.04046	285.03992	1.89	257(40), **241**(100), 217(50), 199(70), 175(70)	255(50), **227**(100), 211(75), 197(35), 183(85)	−
**56**	8.74	**Apigenin 7-*O*-diacetylhexoside isomer 2**	C_25_H_23_O_12_^–^	515.11950	515.11804	2.83	455(20), 270(10), **269**(100), 268(40)	**225**(100), 201(30), 197(20), 183(25), 149(40)	197(100), 183(50), 181(80), 169(30)
**57**	8.80	**Quercetin *^a^***	C_15_H_9_O_7_^−^	301.03538	301.03483	1.83	271(50), 255(20), **179**(100), 151(80), 107(5)	**151**(100)	107(100), 83(10)
**58**	8.84	**6-Methoxyluteolin**	C_16_H_11_O_7_^–^	315.05103	315.05063	1.27	301(20), **300**(100), 166(5)	283(40), 272(70), 255(50), 243(40), **216**(100)	201(25), 188(100), 173(20)
**59**	9.05	**Apigenin 7-*O*-diacetylhexoside isomer 3**	C_25_H_23_O_12_^–^	515.11950	515.11823	2.47	455(20), 293(10), 270(10), **269**(100), 268(20)	**225**(100), 201(25), 197(35), 183(30), 149(40)	210(20), 197(100), 183(30), 181(60), 169(30)
**60**	9.53	**Apigenin 7-*O*-diacetylhexoside isomer 4**	C_25_H_23_O_12_^–^	515.11950	515.11761	3.67	455(30), 413(10), 311(15), 270(15), **269**(100)	**225**(100), 201(30), 197(30), 183(30), 149(35)	207(10), 197(100), 183(30), 181(70), 169(40)
**61**	9.56	**Apigenin *^a^***	C_15_H_9_O_5_^−^	269.04554	269.04449	3.90	225(5), 177(15), **151**(100)	**65**(100)	−
**62**	9.68	**Apigenin 7-*O*-diacetylhexoside isomer 5**	C_25_H_23_O_12_^–^	515.11950	515.11835	2.23	455(10), 270(10), **269**(100), 268(30)	**225**(100), 201(30), 197(25), 183(25), 149(30)	197(70), 183(50), 181(100), 169(30)
**63**	9.73	**Kaempferol *^a^***	C_15_H_9_O_6_^−^	285.04046	285.03969	2.70	**255**(100), 227(10)	**211**(100), 195(5), 167(15)	211(40), 137(100)
**64**	9.81	**Chrysoeriol *^a^***	C_16_H_11_O_6_^–^	299.05611	299.05565	1.54	285(10), **284**(100)	**256**(100)	239(10), 227(100), 212(20), 200(15)
**65**	9.95	**Isorhamnetin**	C_16_H_11_O_7_^–^	315.05103	315.05017	2.73	301(20), **300**(100)	283(30), **271**(100), 255(40), 227(50), 151(90)	243(100), 227(50), 215(10), 199(20)
**66**	10.32	**Chrysosplenol**	C_18_H_15_O_8_^–^	359.07724	359.07703	0.58	345(10), **344**(100), 287(10), 240(5)	**329**(100), 301(5)	314(100), 301(20), 286(30), 270(5), 175(5)
**67**	11.74	**Chrysosplenetin**	C_19_H_17_O_8_^–^	373.09289	373.09277	0.32	359(10), **358**(100)	**343**(100)	328(100), 315(15), 300(30), 284(10), 272(10)

*^a^* Confirmed using available standards; all the other compounds were identified based on HRMS data. **Bold** numbers are peaks which were further fragmented in the MS^3^ and MS^4^ experiments.

**Table 2 plants-12-01211-t002:** Concentration (mg/L) of individual phenolics found in optimal chamomile extracts.

Compound Name	mg/L
**Phenolic acids and their derivatives**	
Gallic acid	0.072
Protocatechuic acid	0.649
*p*-Hydroxybenzoic acid	1.619
Gentisic acid	0.510
Caffeic acid	0.494
*p*-Coumaric acid	0.762
**Flavonoid aglycones and glycosides**	
Quercetin 3-*O*-galactoside	0.283
Isorhamnetin 3-*O*-glucoside	0.569
Apigenin 7-*O*-glucoside	2.408
Naringenin	0.142
Luteolin	0.191
Quercetin	0.861
Apigenin	1.542
Kaempferol	0.358
Chrysoeriol	0.286

**Table 3 plants-12-01211-t003:** Antioxidant potential of optimized MAE chamomile extract.

Parameters	Results
**Antioxidant assays**	
DPPH (mg TE/g)	60.24 ± 2.79
ABTS (mg TE/g)	126.92 ± 9.50
CUPRAC (mg TE/g)	123.12 ± 0.93
FRAP (mg TE/g)	95.37 ± 1.98
MCA (mg EDTAE/g)	21.34 ± 0.06
PBD (mmol TE/g)	1.42 ± 0.06
**Enzyme inhibitory assays**	
AChE inhibition (mg GALAE/g)	0.85 ± 0.11
Amylase inhibition (mmol ACAE/g)	0.18 ± 0.01
Glucosidase inhibition (mmol ACAE/g)	13.11 ± 0.72

Values are reported as mean ± SD of three parallel measurements. TE: Trolox equivalent; EDTAE: EDTA equivalent. GALAE: Galantamine equivalent; ACAE: Acarbose equivalent.

**Table 4 plants-12-01211-t004:** Full factorial design 33 and experimentally observed response.

Inputs			Output
Microwave Power (kW)	Extraction Time (min)	Solid-to-Solvent Ratio	TPC
400	40	1:60	46.04
400	30	1:80	46.16
400	20	1:60	42.85
400	40	1:40	44.18
400	30	1:40	37.99
400	20	1:40	35.54
400	30	1:60	58.435
400	20	1:80	56.75
400	40	1:80	54.14
600	20	1:80	50.68
600	20	1:40	35.95
600	20	1:60	40.65
600	40	1:40	38.47
600	40	1:60	48.5
600	30	1:40	38.07
600	40	1:80	47.46
600	30	1:60	54.51
600	30	1:80	51.86
800	30	1:60	46.9
800	30	1:80	41.6
800	40	1:40	46.29
800	20	1:80	57.32
800	30	1:40	44.34
800	40	1:80	59.56
800	20	1:40	37.17
800	20	1:60	47.88
800	40	1:60	49.36

## Data Availability

Not applicable.

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
