# Peer review of "New Biological and Chemical Insights into Optimization of Chamomile Extracts by Using Artificial Neural Network (ANN) Model"

_plants, 2023, doi:10.3390/plants12061211_

Round 1

Reviewer 1 Report

The author optimized the extract method of Chamomile, then analysis the chemicals and the biological activities, which will be benefit for the development of this plant. Therefore, i suggested this manuscript could be accepted after minor revision. The comments are as below.

1. the parameters of extract method including solid to liquid ratiomicrowave, time was optimized, how about the kinds of solvent (70% ethanol ?) and temperature?

2. What does extracted my ethanol means in line 116?

3. MSn should be MSn.

Author Response

Dear reviewers,

The authors are very grateful for a constructive and objective review and we are grateful for patience and very useful comments which improved the manuscript quality. We made a point by point list of our responses to the reviewer's comments below. All comments and necessary changes suggested by the reviewers were accepted and done. Corrections were made as the reviewer suggested and additional references were included in the manuscript accordingly. Also, some minor technical corrections were made for better understanding and reproduction of work.

Please find below all our point-by-point replies!

Reviewer #1

The author optimized the extract method of Chamomile, then analysis the chemicals and the biological activities, which will be benefit for the development of this plant. Therefore, i suggested this manuscript could be accepted after minor revision. The comments are as below.

Comment #1:

The parameters of extract method including solid to liquid ratio, microwave, time was optimized, how about the kinds of solvent (70% ethanol ?) and temperature?

Response #1: Thanks for the constructive comment.  During the extraction, the choice of solvent and temperature was made on the basis of our earlier research, experience, but also on the basis of liperature. For example, based on the review of the literature, as well as on the basis of our earlier research, ethanol is the most suitable solvent for the extraction of polyphenolic components from chamomile. Ethanol is a solvent that exhibits good solvent properties, easily penetrates plant tissue and has antibacterial properties. By mixing with water in a concentration of 70%, a solvent with significantly better solvent characteristics for isolating polyphenols from chamomile is obtained compared to pure ethanol.

Comment #2:

What does extracted my ethanol means in line 116?

Response #2: Thank you for noticing this omission. The authors made a typo and wrote “my” instead of “by”. We apologize for this oversight. In the revised paper, this omission has been corrected.

Comment #3:

MSn should be MSn.

Response #3:

Corrected as suggested.

Reviewer 2 Report

The authors of this manuscript employed machine learning to optimize the extractions of the dried ligulate flowers of chamomile. The optimization of process parameters was performed with artificial neural networks (ANN) model using solid-to-solvent ratio, microwave power and time as inputs, while the outputs were the yield of total phenolic compounds (TPC). Chemical profiling and bioassays for the optimized extracts were performed. Overall, the discovery of this study is interesting but not impressive. Basically, very similar results can be obtained without the artificial neural networks (ANN) model. Therefore, the manuscript is not suitable for being published in Plants.

Author Response

Dear reviewers,

The authors are very grateful for a constructive and objective review and we are grateful for patience and very useful comments which improved the manuscript quality. We made a point by point list of our responses to the reviewer's comments below. All comments and necessary changes suggested by the reviewers were accepted and done. Corrections were made as the reviewer suggested and additional references were included in the manuscript accordingly. Also, some minor technical corrections were made for better understanding and reproduction of work.

Please find below all our point-by-point replies!

Reviewer #2

The authors of this manuscript employed machine learning to optimize the extractions of the dried ligulate flowers of chamomile. The optimization of process parameters was performed with artificial neural networks (ANN) model using solid-to-solvent ratio, microwave power and time as inputs, while the outputs were the yield of total phenolic compounds (TPC). Chemical profiling and bioassays for the optimized extracts were performed. Overall, the discovery of this study is interesting but not impressive. Basically, very similar results can be obtained without the artificial neural networks (ANN) model. Therefore, the manuscript is not suitable for being published in Plants.

Response:

We are grateful for the comments. Every critical suggestion and comment is valuable and helps us to steer the research in the right direction. We believe that constructive criticism is doing us more good than just positive revision.

Reviewer 3 Report

The manuscript entitled "New biological and chemical insights into optimization of chamomile extracts by using artificial neural networks (ANN) model” has been investigated in detail. The topic addressed in the manuscript is potentially interesting and the manuscript contains some practical meanings, however, there are some issues which should be addressed by the authors:

·         There are several typos errors in Table 1. For example, in solid to solvent ratio column. Please correct it.

·         Please explain more regarding the used ANN model. Which kind of ANN has been used? With which optimization algorithm? Levenberg Marquardt? If yes, please explain in the text.

·         I think the authors uses MLP. If it is true, please explain how the best structure for MLP model has been obtained? For example, how the optimized number of layers, number of neurons in each layer, number of epochs, kind of activation functions and etc. were obtained? Please explain in the text precisely.

·         I recommend the authors to review other recently developed works.

·         Do the authors check reproducibility for experiments? Please add to the text.

·         There is a regression diagram in Fig. 1. Please change it with 2 regression diagrams for both training and testing sets. The readers can find many things from these separated graphs.  

·         I think the range of difference between ANN inputs is not ideal but acceptable. Please consider it in future researches.

·         Was the underfitting and overfitting problem occurred? Please explain about it.  

·         Other error parameters such as MRE and RMSE could be used.

This study may be consider for publication if it is addressed in the specified problems.

Author Response

Dear reviewers,

The authors are very grateful for a constructive and objective review and we are grateful for patience and very useful comments which improved the manuscript quality. We made a point by point list of our responses to the reviewer's comments below. All comments and necessary changes suggested by the reviewers were accepted and done. Corrections were made as the reviewer suggested and additional references were included in the manuscript accordingly. Also, some minor technical corrections were made for better understanding and reproduction of work.

Please find below all our point-by-point replies!

Reviewer #3

The manuscript entitled "New biological and chemical insights into optimization of chamomile extracts by using artificial neural networks (ANN) model” has been investigated in detail. The topic addressed in the manuscript is potentially interesting and the manuscript contains some practical meanings, however, there are some issues which should be addressed by the authors:

Comment #1:

There are several typos errors in Table 1. For example, in solid to solvent ratio column. Please correct it.

Response #1:

We are sorry for typing errors, corrected as suggested.

Comment #2:

Please explain more regarding the used ANN model. Which kind of ANN has been used? With which optimization algorithm? Levenberg Marquardt? If yes, please explain in the text.

Response #2:

We are sorry for misunderstanding. The architecture of the developed artificial neural network is explained in the 2.3 section (Lines 139-144).

Comment #3:

I think the authors uses MLP. If it is true, please explain how the best structure for MLP model has been obtained? For example, how the optimized number of layers, number of neurons in each layer, number of epochs, kind of activation functions and etc. were obtained? Please explain in the text precisely. I recommend the authors to review other recently developed works.

Response #3:

Developed artificial neural network (ANN) has a classical multilayer perceptions model. Obtaining the best ANN architecture was achieved by a trial-and-error iteration process. The criteria to stop iterations were the highest value of the coefficient of determination and the lowest values for the mean absolute value (MAE). Moreover, we have added the latest work by the group from our faculty that also successfully use a similar ANN configuration to predict extracted polyphenols from peppermint. Added text can be found in lines 237-239, and the appropriate reference has been added to the list.

Comment #4:

Do the authors check reproducibility for experiments? Please add to the text.

Response #4:

In order to experimentally validate the optimal solutions obtained by ANN and particle swarm algorithm, the TPC extraction was repeated at obtained optimal conditions and the maximum absolute relative error was smaller than 5%.

Comment #5:

There is a regression diagram in Fig. . Please change it with 2 regression diagrams for both training and testing sets. The readers can find many things from these separated graphs. 

Response #5:

The Figure has been divided in two parts that present training and test data.

Comment #6:

I think the range of difference between ANN inputs is not ideal but acceptable. Please consider it in future researches.

Response #6:

Thank you for the suggestion. The range of experimental data used for the analysis and for ANN inputs were derived using the full factorial design of experiments. The design was used to limit the sample size to a value of 27 experiments which is sufficient for the developing topology of the ANN.

Comment #7:

Was the underfitting and overfitting problem occurred? Please explain about it. 

Response #7:

Thank you for your valuable suggestions, we have had in mind overfitting problems during developing topologies of the ANN. ANN results and fitting problems, including the weight values, depend on the initial assumptions of parameters necessary for ANN construction and fitting. In the same way, the different number of hidden neurons can give different ANN outcomes. In this context, a series of different topologies were used, in which the number of hidden neurons were varied from 1 to 20 and the training process of each network was run ten times with random initial values of weights and biases. The creation of 200 ANN in total was the result of this procedure. During the iteration process of changing ANN topologies coefficient of determination and mean absolute error became constant and that was verification to choose the best ANN without the worry of overfitting and reaching local optima.

Comment #8:

Other error parameters such as MRE and RMSE could be used. This study may be consider for publication if it is addressed in the specified problems.

Response #8:

Developed ANN successfully predict experimental data, verification can be seen from Figure 1 and MAE and R2, so there was no need for additional error models criteria such as MRE and RMSE.

Author Response

Dear reviewers,

The authors are very grateful for a constructive and objective review and we are grateful for patience and very useful comments which improved the manuscript quality. We made a point by point list of our responses to the reviewer's comments below. All comments and necessary changes suggested by the reviewers were accepted and done. Corrections were made as the reviewer suggested and additional references were included in the manuscript accordingly. Also, some minor technical corrections were made for better understanding and reproduction of work.

Please find below all our point-by-point replies!

Reviewer #4

In the paper „New biological and chemical insights into optimization of chamomile extracts by using artificial neural networks (ANN)” the authors performed ethanol and microwave extraction of the plant material and optimized this procedure by applying the ANN methodology. Very interesting scientific work, but it needs some corrections:

Comment #1:

Line 55 and 59 repetition of the same information

Response #1:

The authors are grateful for this remark. In revised version of the manuscript the mentioned remark has been corrected.

Comment #2:

line 63 remove the parenthesis.

Response #2:

We apologies. Corrected as suggested.

Comment #3:

Line 116 my ethanol ?

Response #3: Thank you for noticing this omission. The authors made a typo and wrote “my” instead of “by”. We apologize for this oversight. In the revised paper, this omission has been corrected.

Comment #4:

table 2 is not readable, it is necessary to correct table 2

Response #4:

Thank you for your observation. Table 2 contains a lot of data regarding the identification of even 67 different compounds. Forming this table and obtaining data for it required a lot of attention, observation, and perseverance. In an effort to make the table as readable as possible, the authors made a couple of technical changes. However, due to its size and the amount of data, we were limited. In addition, the authors noticed that in the word system in which we originally created the manuscript, the table is clearer and more readable than when we download the document. Therefore, if the work is accepted, additional technical processing will be done before its publication, so that all data from the table will be clear and readable.

Reviewer 5 Report

Review – manuscript no. plants-2086119

Review of the manuscript which has been submitted to Plants

Manuscript no. plants-2086119

In the current context of the study topic, the article entitled “New biological and chemical insights into optimization of chamomile extracts by using artificial neural networks (ANN) model” is well chosen and interesting. The methodology is very complex and very well described. I greatly appreciate the titanic work done to identify the polyphenols in the MS spectra, a painstaking work that requires patience and perseverance. To draw up a table like Table 2 requires a lot of attention, observation, and perseverance, let alone calculating and identifying 67 compounds from all the m/z in the spectra. Well done! The article is very well written but still below I have some observations and I made some suggestions to improve the quality of the work in order to recommend acceptance for publication.

Page 3, row 119: how the domestic microwave oven was modified?

Page 5, row 183,: please detail the abbreviation “LA”

Page 5, rows 188: please correct the fermentation degree by adding the correct sign;

Page 13, row 374: you can also complete studies related to the fermentation of the lactic acid bacteria in a media containing vegetal extracts with the following work: https://doi.org/10.3390/fermentation8100553.

Author Response

Dear reviewers,

The authors are very grateful for a constructive and objective review and we are grateful for patience and very useful comments which improved the manuscript quality. We made a point by point list of our responses to the reviewer's comments below. All comments and necessary changes suggested by the reviewers were accepted and done. Corrections were made as the reviewer suggested and additional references were included in the manuscript accordingly. Also, some minor technical corrections were made for better understanding and reproduction of work.

Please find below all our point-by-point replies!

Reviewer #5

In the current context of the study topic, the article entitled “New biological and chemical insights into optimization of chamomile extracts by using artificial neural networks (ANN) model” is well chosen and interesting. The methodology is very complex and very well described. I greatly appreciate the titanic work done to identify the polyphenols in the MS spectra, a painstaking work that requires patience and perseverance. To draw up a table like Table 2 requires a lot of attention, observation, and perseverance, let alone calculating and identifying 67 compounds from all the m/z in the spectra. Well done! The article is very well written but still below I have some observations and I made some suggestions to improve the quality of the work in order to recommend acceptance for publication.

Comment #1: Page 3, row 119: how the domestic microwave oven was modified?

Response #1: The microwave oven was modified by adding a condenser which was connected to the reaction vessel (which contained the mixture of solvent and plant). In order to make it clearer for the readers, in the revised version of the manuscript, a picture of the modified oven was added.

Comment #2: Page 5, row 183: please detail the abbreviation “LA”

Response #2: Corrected as suggested, the explanation of LA abbreviation was added (Page 5, line 187).

Comment #3: Page 5, rows 188: please correct the fermentation degree by adding the correct sign;

Response #3: Corrected as suggested.

Comment #4: Page 13, row 374: you can also complete studies related to the fermentation of the lactic acid bacteria in a media containing vegetal extracts with the following work: https://doi.org/10.3390/fermentation8100553.

Response #4: Thank you for the suggestion. The following text was added to the discussion: “Similar results were observed by Dimofte et al. [58] when fermentation media for Weissella confuse PP29, was enriched with deferment concentrations of anthocyanins from Hibiscus sabdariffa L. The fermentation media rich in anthocyanins (high concentrations) stimulated the growth of Weissella confuse and biosynthesis of exopolysaccharide, thus leading to augmented probiotic and prebiotic properties.“

Also the reference has been cited in Reference section.

Round 2

Reviewer 2 Report

Thanks for the authors' efforts for the revision. The paper is still lack of novelty for being published in Plants. 

Author Response

Reviewer 2: Thanks for the authors' efforts for the revision. The paper is still lack of novelty for being published in Plants. 

Response:  The authors are grateful for the remark and suggestion. Nevertheless, we believe that this work provides certain novelties, and contains data that cannot be found in the literature until now.

It is worth noting that scientists have already tried to optimize complex extraction processes with many mathematical models. Our contribution reflects on the fact that for the first time the advanced methodology that combines Artificial Neural Networks (ANNs) and the Particle swarm algorithm is used in order to optimize the nonlinear and complex process of extraction dried ligulate flowers of chamomile. Artificial Neural Networks are novel compared to traditional models. ANNs can learn and adapt to new information through training, whereas traditional models rely on explicit programming and assumptions about the relationship between inputs and outputs. This structure allows ANNs to handle complex and non-linear relationships between inputs and outputs, which is a challenge for traditional models such as polynomials, splines and response surface methodology. We have proved this on the wide range of technological processes in the following papers, Microwave-assisted extraction of peppermint polyphenols – Artificial neural networks approach. Food and Bioproducts Processing, (2019) 118, 258–269. https://doi.org/https://doi.org/10.1016/j.fbp.2019.09.016 and Multiobjective process optimization for betaine enriched spelt flour based extrudates. Journal of Food Process Engineering, 2019, 42(1), https://doi.org/10.1111/jfpe.12942 .

Overall, the novel structure and capabilities of ANNs allow them to tackle complex problems and produce state-of-the-art results in many applications, making them a valuable tool in modern machine learning and artificial intelligence.

Moreover, used Particle Swarm Optimization (PSO) is a computational optimization technique that has several advantages compared to other local optimization techniques, including that PSO is relatively easy to understand and implement compared to other optimization algorithms, making it accessible to a wide range of users and data. Therefore, other investigators can easily reuse it on their experimental data. One of the most important advantages is that PSO has the ability to search for global optimal solutions, making it well-suited for optimization problems with multiple, complex objectives. Also, it can be applied to a wide range of optimization problems, including continuous and discrete optimization problems. Moreover, PSO is relatively robust to the initial starting points, meaning it can converge to a global optimum even if the starting point is not near the optimum.

In addition to this, we strongly believe that the work provides new information regarding the biological activity of the plant material prepared in the way different from the usual procedure. Namely, the paper used fermented plant material (by endogenous plant enzymes activated by optimal pH using sodium-acetate buffer), while other works deal with pure chamomile. We examined fermented plant because, according to our earlier knowledge, fermented material shows a higher degree of activity and greater potential for application in industry (pharmaceutical, food and cosmetic). By reviewing the literature, we determined that there are no data on the enzyme-inhibitory effect of fermented chamomile flowers, and that their antioxidant power has been very superficially tested. That's why we applied a set of even 6 different assays in order to obtain a more complete picture and provide a real insight into the antioxidant capacity of fermented chamomile, and therefore show its potential for the industrial application.

Finally, in our study, we have used the advanced novel techniques which can be easily reproduced by other researchers and which are necessary to precisely predict and obtain the optimal solution of extraction processes.

Reviewer 3 Report

All of the comments have been applied in the manuscript so my recommendation is "Accept".

Author Response

Reviewer 3: All of the comments have been applied in the manuscript so my recommendation is "Accept".

Response: Thank you for all the previous advice and constructive criticism thanks to which we improved the quality of the manuscript. Finally, we would like to thank you for your positive decision.
